# Preterm Birth and SARS-CoV-2: Does a Correlation Exist?

**DOI:** 10.3390/biomedicines13020282

**Published:** 2025-01-24

**Authors:** Federica Perelli, Annalisa Vidiri, Giovanna Palomba, Rita Franco, Vitalba Gallitelli, Marco Parasiliti, Marta Bisanti, Amelia Spanò, Adele Silvagni, Alessandra Lopez, Giuseppe Gullo, Gaspare Cucinella, Anna Franca Cavaliere

**Affiliations:** 1Pediatric Gynecology Unit, Meyer Children’s Hospital IRCCS, 50139 Florence, Italy; federica.perelli@meyer.it; 2Azienda USL Toscana Centro, Gynecology and Obstetrics Department, Santa Maria Annunziata Hospital, 50012 Florence, Italy; 3Obstetrics and Gynecology Cannizzaro Hospital, Kore University of Enna, 95126 Catania, Italy; 4Division of Gynecology and Obstetrics, IRCSS Azienda Ospedaliera Universitaria of Bologna, 40138 Bologna, Italy; giovannapalomba@gmail.com; 5Division of Gynecology and Obstetrics, Isola Tiberina Gemelli Hospital, 00186 Rome, Italy; rita.franco.fw@fbf-isola.it (R.F.); vitalba.gallitelli.fw@fbf-isola.it (V.G.); marco.parasiliti@fbf-isola.it (M.P.); 6Department of Science of Woman, Child and Public Health, Fondazione Policlinico Universitario A. Gemelli IRCCS, Università Cattolica del Sacro Cuore, 00168 Rome, Italy; bisantimarta.93@gmail.com (M.B.); ameliaspano94@gmail.com (A.S.); 7Research Unit of Gynaecology, Department of Medicine and Surgery, Università Campus Biomedico, 00128 Rome, Italy; adelesilvagni95@gmail.com; 8Unit of Gynecology and Obstetrics, AOOR Villa Sofia Cervello, University of Palermo, 90146 Palermo, Italy; alessandralopez91@gmail.com (A.L.); gullogiuseppe@libero.it (G.G.); gaspare.cucinella1@gmail.com (G.C.); 9Fondazione Policlinico Universitario Agostino Gemelli IRCCS, 00136 Rome, Italy; afcavaliere@hotmail.com

**Keywords:** SARS-CoV-2, COVID-19, preterm birth, pregnancy outcomes, obstetric outcomes

## Abstract

**Introduction:** The emergence of the SARS-CoV-2 virus and its subsequent global pandemic have raised significant concerns regarding its impact on pregnancy outcomes. This review aims to summarize the emerging data on the risk of preterm delivery in pregnant women infected with SARS-CoV-2. **Materials and Methods:** A systematic search was conducted from March 2020 to December 2023 using PubMed and Web of Science, following PRISMA guidelines. Studies correlating maternal COVID-19 infection with preterm birth were included. **Results:** Thirteen studies were analyzed, indicating a higher incidence of preterm birth in SARS-CoV-2-positive pregnant women compared to controls. The average incidence rate of preterm birth in infected patients was 18.5%, with a median of 12.75%, while non-infected women showed an average incidence of preterm birth of 10%, with a median of 8.2%. **Discussion:** Studies suggest an association between SARS-CoV-2 infection during pregnancy and increased risk of preterm birth and cesarean section. The severity of symptoms and underlying comorbidities further elevate this risk. Notably, infections during the third trimester pose the highest risk of preterm birth. **Conclusion:** Preventing SARS-CoV-2 infection during pregnancy is crucial to mitigate adverse obstetric outcomes. Close monitoring and tailored interventions for infected pregnant women, particularly those in later trimesters and with comorbidities, are imperative to reduce the risk of preterm birth and improve maternal-fetal outcomes.

## 1. Introduction

SARS-CoV-2 infection gave rise to a global pandemic in 2020, infecting about 700 million people of all age groups and causing approximately 6.5 million deaths [1]. The infected population is evenly distributed across age groups, with a significant involvement of pregnant women. Reports from various research teams indicate an increase in preterm deliveries among pregnant women infected by the virus. Specifically, the incidence of preterm births (PTB) in infected pregnant women has been reported to be between 14% and 25%, particularly in high-income countries [2,3,4,5,6].

One of the primary factors contributing to the higher incidence of preterm delivery in infected women is the occurrence of severe maternal respiratory infections, necessitating the early extraction of the fetus to safeguard maternal health [7,8,9]. Additionally, a retrospective analysis conducted in France highlighted a notable incidence of spontaneous preterm deliveries in pregnant women infected with SARS-CoV-2 [10].

The objective of our review was to summarize emerging data on the risk of preterm delivery in pregnant women with SARS-CoV-2 infection. An online search was conducted from 20 March 2020 to 8 December 2023 to identify studies regarding pregnancy outcomes in women affected by the virus. The search strategy followed PRISMA guidelines to ensure a comprehensive review of published studies. By providing a thorough analysis of the current literature, we advance the understanding of SARS-CoV-2’s impact on pregnancy outcomes, particularly preterm birth. Our findings have significant implications for clinical practice, emphasizing the need for heightened awareness and monitoring of pregnant women infected with this virus.

## 2. Materials and Methods

This search was performed on PubMed and Web of Science using the keywords “preterm birth”, “COVID-19 infection”, “SARS-CoV-2 virus”, “pregnancy”, and “maternal health”. This search included all articles published from March 2020 to December 2023 that fulfilled specific eligibility criteria. The criteria included are as follows:Population: Studies must involve pregnant women diagnosed with SARS-CoV-2 infection;Intervention/Exposure: Research focusing on the correlation between maternal COVID-19 infection and preterm birth (PTB) outcomes;Outcomes: Studies must provide data on the incidence of PTB, defined as birth before 37 weeks of gestation, and should categorize PTB as either spontaneous or medically indicated;Study Design: Included study designs were case-control studies, comparative studies, retrospective cohort studies, cross-sectional studies, and prospective observational studies;Language: Only articles written in English were included;Quality and Relevance: Studies must present original data and avoid redundancy by excluding review articles, meta-analyses, or studies solely investigating the incidence of PTB during the COVID-19 era without considering the direct correlation between SARS-CoV-2 infection and PTB. Review and meta-analysis articles were excluded to mitigate the risk of including the same study multiple times in this review.

The protocol for this review was created in accordance with the Preferred Reporting Items for Systematic Review and Meta-Analysis Protocols (PRISMA-P) guidelines. The study protocol was registered in the records of the Open Science Framework (https://osf.io/4mzcp, accessed on 11 December 2024). The selection process of eligible studies is illustrated in Figure 1. Three independent reviewers evaluated the titles and abstracts of the identified articles. From an initial pool of 315 articles (282 from Pubmed and 33 from Web of Science), 13 were selected as they met the inclusion criteria. Only those that complied with the following conditions were included: case-control studies; comparative studies; retrospective cohort studies; cross-sectional studies; and prospective observational studies, all conducted on human subjects and written in English. Included studies specifically measured exposure to SARS-CoV-2 infection using rapid antigen tests or molecular polymerase chain reaction (PCR) tests, with PTB designated as either a primary or secondary outcome. Articles solely investigating the incidence of PTB during the COVID-19 era were excluded since our aim was to explore the exposure–outcome correlation. Preterm birth rates in pregnant women with COVID-19 in included studies are reported in Figure 2.

Data extraction was independently performed by three investigators (G.P., A.S., M.B.). Specifically, data concerning the prevalence of PTB among women with SARS-CoV-2 infection were compared to that in control groups. All reviewed studies based their diagnosis of SARS-CoV-2 infection on PCR or antigen test results. Moreover, PTB was defined as occurring before 37 weeks of gestation, in accordance with World Health Organization (WHO) criteria. A correlation study was also undertaken to calculate the Pearson coefficient.

## 3. Results

Thirteen studies were included in this review, examining the incidence of preterm birth (PTB) among pregnant patients with SARS-CoV-2 infection (confirmed by PCR or antigen tests). The findings indicate that the average incidence rate of PTB among infected patients is 18.5%, with a median of 12.75%. In contrast, control groups (women without SARS-CoV-2 infection) exhibited an average PTB incidence of 10%, with a median of 8.2%. Most studies demonstrate that women infected with SARS-CoV-2 are more likely to experience PTB compared to those who are uninfected.

Several limitations were noted, including the possibility of false negatives in asymptomatic control groups, small sample sizes in some cases, and the exclusion of stillbirths in certain studies, which may underrepresent adverse outcomes. Nevertheless, results were consistent across different geographical areas and socio-economic backgrounds. For instance, a study in Mumbai, India, highlighted a high PTB rate and increased maternal complications requiring ICU admission among COVID-19-infected pregnancies [11]. A similar finding was confirmed in a multicenter case-control study in the United States [12].

Many studies indicate that the higher PTB rate in SARS-CoV-2 infected patients is often due to medically indicated deliveries prompted by unfavorable maternal or fetal conditions. Ferrara et al. reported that 62.24% of PTBs in infected patients were induced compared to 37.76% that were spontaneous [13]. Furthermore, research shows that severe COVID-19 or related complications have a significant impact on the incidence of iatrogenic PTB. Bahado et al. noted that many iatrogenic PTBs in the control group were due to preeclampsia, particularly among symptomatic COVID-19 patients (11%) compared to 7.4% for other causes in infected patients [12].

Additionally, Smith et al. [14] found that symptomatic COVID-19 cases had a higher risk of iatrogenic PTB compared to mild cases, while asymptomatic infections had rates similar to uninfected women. The overall iatrogenic PTB rate in their study cohort was 38%, compared to 61.84% for spontaneous PTB, suggesting that SARS-CoV-2 infection itself may trigger spontaneous labor or premature rupture of membranes (pPROM).

Common symptoms necessitating iatrogenic deliveries included respiratory distress, severe pulmonary issues, and other serious conditions [14,15,23,24,25], highlighting that severe COVID-19 late in pregnancy markedly raises the risk of PTB, mainly through medically indicated deliveries but could also influence spontaneous PTB rates [13].

Notably, Bahado et al. [12] found a dose-response relationship between COVID-19 severity and earlier delivery. Higher COVID-19 severity was associated with higher rates of medically indicated PTB, primarily due to preeclampsia.

Some studies identified that the risk of PTB is highest when SARS-CoV-2 infection occurs in the third trimester, especially within the initial four days post-infection [16,26]. Darling et al. reported increased provider-initiated PTB risk during the second trimester, while the risk for both PTB types was heightened during the third trimester [17]. Furthermore, Bobei et al. highlighted that elevated C-reactive protein (CRP) and low lymphocyte counts are potential risk factors for PTB in SARS-CoV-2-positive pregnant women [16].

Pregnant women with SARS-CoV-2 infection also showed a higher likelihood of requiring cesarean sections, particularly in cases of fetal distress or severe maternal complications. Studies confirmed that symptomatic women with COVID-19 were more frequently subjected to cesarean deliveries [9,16,18,19,24,27]. One study reported a cesarean rate of 82% among SARS-CoV-2-positive patients, compared to only 6% in controls, with many cesarean sections attributed to fetal distress or worsening maternal symptoms [15].

Similar trends were observed in Epelboin’s study [10], which documented cesarean rates of 33% in infected women versus 20.2% in the control group. Metz et al. [23] corroborated this increase in cesarean rates, indicating 59.6% among COVID-19-positive women compared to 34% in the control group (adjusted risk ratio of 1.57). Additionally, a multinational cohort study by Giuliani et al. [24] demonstrated that newborns born via cesarean delivery to SARS-CoV-2-positive mothers were at greater risk of contracting the virus, emphasizing the potential risks associated with cesarean delivery under such circumstances.

Neonatal outcomes varied across studies. Mahajan et al. [11] found no notable differences in Apgar scores or birth weights between this study and control groups, while Trahan’s study [24] similarly found no significant differences in obstetrical and newborn outcomes between infected and uninfected patients. Conversely, Bobei’s study [16] indicated that severe maternal symptoms, such as respiratory distress, negatively influenced newborn outcomes (*p* = 0.001). Additionally, Giuliani’s research [24] showed a direct correlation between the length of in utero exposure to COVID-19 and the likelihood of newborns testing positive for the virus. When newborns from mothers who tested positive also contracted the virus, they faced worse outcomes, including elevated rates of NICU admissions, fever, gastrointestinal and respiratory symptoms, and even death (adjusted for prematurity). Respiratory symptoms and NICU admissions were more frequently observed in newborns of mothers diagnosed with COVID-19.

Vouga et al. [20] noted an 8–11% rate of severe outcomes in infected pregnant women, exceeding that of the general population in analogous age groups. Factors such as vascular and hemodynamic changes, along with altered immunity and compromised respiratory function in pregnant women, play a significant role in these outcomes. Additionally, the potential for vertical transmission from mother to fetus or newborn raises concerns, particularly given instances of placental infection that could jeopardize the unborn child’s health.

The study by Blitz et al. examined PTB rates among women with either symptomatic or asymptomatic SARS-CoV-2 infections. After adjusting for maternal age, race-ethnicity, parity, prior PTB history, body mass index, marital status, comorbidities, delivery month, and pandemic wave, it was observed that patients with symptomatic COVID-19 at the time of delivery had a notably higher risk of PTB (19.0%) compared to those with asymptomatic infections (8.8%) or no infections (7.1%) [21].

Lastly, Gulersen et al. [22] found that during the Omicron wave, pregnant women with SARS-CoV-2 infection faced an increased risk of PTB compared to uninfected cases. This risk was not influenced by vaccination status or the presence of SARS-CoV-2 antibodies, as evidenced by subgroup analyses. While the association between SARS-CoV-2 infection and PTB was well established during the pandemic’s early stages, this risk persisted even during later, less virulent phases. These findings reaffirm that SARS-CoV-2 infection in pregnancy represents a significant risk factor for PTB, highlighting the dangers posed by both current and emerging variants [22].

Collected data from the included studies are reported in Table 1.

When interpreting the findings of this review, it is crucial to address the potential biases inherent in the 13 included studies, each characterized by diverse methodologies and observational designs. The studies encompass a range of types—ranging from large-scale retrospective cohort studies to smaller case-control and cross-sectional analyses—which can introduce variability in outcomes related to PTB associated with SARS-CoV-2 infection.

One significant bias arises from the variation in study designs. For instance, retrospective cohort studies, such as those conducted by Epelboin et al. and Darling et al., rely on previously collected data, which may be susceptible to recall bias or incomplete records. This approach can lead to inaccuracies in reporting the prevalence of COVID-19 and associated maternal outcomes. Furthermore, the reliance on clinical diagnoses recorded in medical records may overlook asymptomatic cases of SARS-CoV-2 infection, potentially underrepresenting the true incidence of infection and its effects on pregnancy.

In addition, the included studies vary in their sample sizes and population characteristics. Some studies, such as that by Mahajan et al., feature large cohorts, which can enhance the statistical power and generalizability of findings. However, smaller studies, like those conducted in Bangladesh, may yield results that are less reliable and could reflect unique local factors rather than broader trends. The disparity in patient populations, including geographic, socio-economic, and healthcare access differences, can also introduce confounding variables that impact the association between COVID-19 infection and PTB rates.

Moreover, the studies collectively demonstrate a range of reported PTB rates across different contexts, suggesting that the definitions of PTB and the inclusion criteria for study participants are not consistently applied. For example, some studies, like those by Smith et al. and Bobei et al., assess both spontaneous and medically indicated PTBs, while others may focus exclusively on one type. This inconsistency complicates direct comparisons and increases the risk of misinterpretation regarding the relationship between the severity of COVID-19 and PTB outcomes.

Another noteworthy aspect is the potential for biases introduced by patient selection. The studies conducted in specialized COVID-19 centers may comprise populations with more severe infections, thus skewing the data toward higher rates of adverse outcomes compared to general maternity settings. Additionally, factors such as maternal age, preexisting conditions, and healthcare disparities are often inadequately controlled in these studies, leading to potential confounding effects that may misrepresent the actual impact of SARS-CoV-2 infection on PTB.

To address these biases, a sensitivity analysis is recommended to determine the robustness of the findings. Such an analysis would allow for the examination of how the results might change with the inclusion or exclusion of smaller studies, as well as those with different methodologies. Understanding the impact of these biases will be critical in evaluating the reliability of the conclusions drawn in this review and in informing future research directions.

The different methodologies and inherent biases present in the included studies signify the need for caution when interpreting the relationship between SARS-CoV-2 infection and preterm birth. Future research efforts should strive for greater methodological standardization, clearer definitions of outcomes, and comprehensive consideration of confounding factors to enhance the validity and reliability of findings regarding maternal health in the context of COVID-19.

While the data collectively highlight a concerning association between SARS-CoV-2 infection during pregnancy and increased PTB rates, critical analysis of the strengths and weaknesses of these studies is vital for understanding the implications of the findings. The strengths of the present review are the following:Diverse Populations: The included studies encompassed diverse populations, including various geographical regions and healthcare settings. This diversity enhances the generalizability of the findings across different demographics;Robust Sample Sizes: Many studies included substantial sample sizes, which increased the power of the findings and reduced the likelihood of random error. Larger populations tend to provide a more compelling case for statistically significant associations between infection and outcomes;Use of Established Diagnostic Tests: The studies employed reliable diagnostic methods, such as PCR and antigen tests, to confirm SARS-CoV-2 infection, providing confidence in the classification of subjects as infected or non-infected.

The weaknesses of the present review are the following:Inconsistent Methodologies: A lack of standardized definitions and methodologies among studies may introduce variability in findings. For instance, variations in the classification of PTB (spontaneous vs. medically indicated) can complicate comparisons and meta-analyses;Potential Biases: Some studies used retrospective designs, which can introduce recall and selection bias. The reliance on existing medical records may overlook cases of asymptomatic infections or underreport complications;Exclusion of Key Variables: Several studies did not account for confounding variables such as socio-economic status, prenatal care quality, and the presence of other maternal comorbidities. These factors are essential in understanding the full impact of SARS-CoV-2 infection on PTB;Limited Longitudinal Data: Most studies reported outcomes at single points in time, limiting the ability to assess the long-term impacts of SARS-CoV-2 infection during pregnancy.

In summary, the data from these studies collectively suggest that symptomatic SARS-CoV-2 infection during pregnancy is associated with increased maternal morbidity and adverse maternal outcomes. Infected women are at a higher risk for ICU admission and are susceptible to developing complications such as preeclampsia, hypertensive disorders, coagulation disorders, disseminated intravascular coagulation (DIC), respiratory failure, shock, and organ failure [11,12,15,24,27].

## 4. Discussion

This review addresses two critical questions regarding pregnant women and COVID-19: the associated risks for expectant mothers and whether these risks are influenced by symptom severity or underlying health conditions. The goal was to determine if COVID-19 correlated with adverse obstetric outcomes, specifically increased rates of spontaneous or induced preterm birth (PTB) and cesarean sections.

Numerous studies have explored the relationship between PTB rates and SARS-CoV-2 infection during pregnancy, finding a strong association between severe maternal illness and induced PTB [28]. Most investigations confirm that symptomatic COVID-19 patients face a higher risk of PTB and cesarean delivery compared to asymptomatic individuals, linking symptom severity to outcomes through potential organ damage, primarily vascular and respiratory [14,17]. Specifically, preeclampsia emerged as a leading reason for cesarean sections and medically induced PTB, while fetal compromise and respiratory issues were secondary factors [1,26].

The mechanisms by which SARS-CoV-2 infection may contribute to preterm labor are multifaceted and involve several pathways. One prominent mechanism is the robust inflammatory response triggered by the infection, characterized by a release of cytokines. This cytokine storm can result in systemic inflammation, which has been linked with preterm labor. Elevated levels of pro-inflammatory cytokines, such as interleukin-6 (IL-6), can alter the uterine environment, potentially leading to contractions and cervical changes that initiate premature labor.

Furthermore, COVID-19 has been shown to cause endothelial dysfunction, which may exacerbate preexisting conditions such as preeclampsia. This dysfunction can lead to impaired placental blood flow, resulting in ischemia and placental insufficiency, thereby increasing the risk of preterm birth. Additionally, vascular changes associated with COVID-19 could contribute to the development of thromboembolic events, which complicate maternal health and may necessitate early delivery.

Severe respiratory symptoms from COVID-19 can also induce hypoxia, depriving both the mother and fetus of vital oxygen. These hypoxic conditions can adversely affect fetal development and may trigger pathways leading to preterm birth, as studies indicate that maternal hypoxia can induce labor through mechanisms involving cervical remodeling and increased uterine contractility.

Moreover, SARS-CoV-2 infection has significant implications for preexisting maternal health conditions, such as obesity, diabetes, and hypertension. The interaction between the virus and these comorbidities often results in a heightened inflammatory state and various vascular complications, further increasing the likelihood of preterm delivery. Additionally, the infection can disrupt normal hormonal balances critical for maintaining pregnancy, leading to changes in levels of hormones such as progesterone and oxytocin. These hormonal alterations may precipitate uterine contractions and cervical dilation, contributing to the onset of labor.

A 2023 retrospective study in Germany analyzing 6086 deliveries revealed a reduction in spontaneous PTB during the pandemic compared to pre-pandemic rates (OR 0.78), alongside an increase in elective and unplanned cesarean sections among term newborns [29]. Another observational study in Romania noted that lockdown measures coincided with increased stillbirths and decreased premature births, raising concerns about the impact of pandemic policies on the identification and treatment of high-risk pregnancies [30].

Pregnant women with multiple comorbidities face heightened risks of PTB and cesarean delivery, with factors such as advanced maternal age, obesity, chronic hypertension, high cholesterol, and preexisting diabetes linked to severe COVID-19 outcomes [19]. The highest incidence of PTB appears within the first thirty-four days following a positive SARS-CoV-2 test, highlighting the importance of lymphocyte counts and inflammation markers, like C-reactive protein (CRP), as risk factors for PTB.

Research from Massachusetts corroborates these findings, showing elevated risk ratios for PTB based on infection timing. Infection during the second trimester posed a higher risk for medically induced PTB, while third-trimester infections influenced both spontaneous and induced PTB rates comparably.

While the risk of spontaneous PTB due to COVID-19 merits attention, it is crucial not to overlook its occurrence; this population sees around twice the incidence of premature rupture of membranes (PROM) compared to the general population [31]. Adverse effects from COVID-19 may likely be linked to significant immune dysregulation, as indicated by altered inflammatory responses and elevated interleukins, which could also contribute to preeclampsia [32].

Preventative measures, especially vaccination, are essential to mitigate serious COVID-19 complications in pregnant women. Recent reviews have confirmed that COVID-19 vaccination does not significantly increase the risk of PTB, reinforcing the need for public health strategies to promote vaccination among pregnant individuals [33,34].

Conflicting guidelines exist regarding the use of antiviral drugs during pregnancy. More research is needed to establish whether fetal-safe therapies can effectively prevent adverse outcomes associated with SARS-CoV-2 infection. Additionally, evidence shows potential for vertical transmission of the virus, highlighting concerns about placental damage and unfavorable obstetric outcomes [6,18,19,26]. However, many studies found that the risk of placental complications and vertical transmission is low, reassuring expecting mothers regarding the virus’s impact on fetal health [35,36,37,38,39,40]. These characteristics differ from other viral infections that are spreading during recent years, such as Zika virus infection [41].

In conclusion, while SARS-CoV-2 infection does not significantly compromise placental or fetal outcomes overall, the incidence of placental pathologies and vertical transmission remains rare. Continuous monitoring of pregnant women, particularly those infected during the later stages of pregnancy or who have significant risk factors, is crucial. Recognizing and managing maternal comorbidities, cytokine dysregulation, and placental health can guide targeted interventions and tailored antenatal care strategies.

## 5. Conclusions

This review underscores the necessity of preventing SARS-CoV-2 infection during pregnancy to mitigate related adverse outcomes. Pregnant women who contract the virus, particularly in the second or third trimester, require close monitoring and careful management of their infection to limit the risks of both spontaneous and induced PTB. Understanding the interplay of maternal health factors, inflammatory responses, and placental conditions will be vital in developing effective care strategies.

Looking to the future, it is imperative that research continues to explore the relationship between SARS-CoV-2 infection and fertility. Investigating how the virus may impact reproductive health, including potential effects on ovulation, implantation, and overall fertility in both men and women, is crucial. Additionally, studies should examine whether COVID-19 vaccination influences fertility outcomes, including understanding any potential effects on sperm quality or oocyte function.

Future research should also focus on elucidating the long-term consequences of COVID-19 on reproductive health, ensuring comprehensive care for individuals planning pregnancies in the aftermath of the pandemic. As more data become available, it will be important to inform clinical practices and public health recommendations to support the reproductive health of those of childbearing age and ensure safe pregnancies amid ongoing concerns related to COVID-19.

## Figures and Tables

**Figure 1 biomedicines-13-00282-f001:**
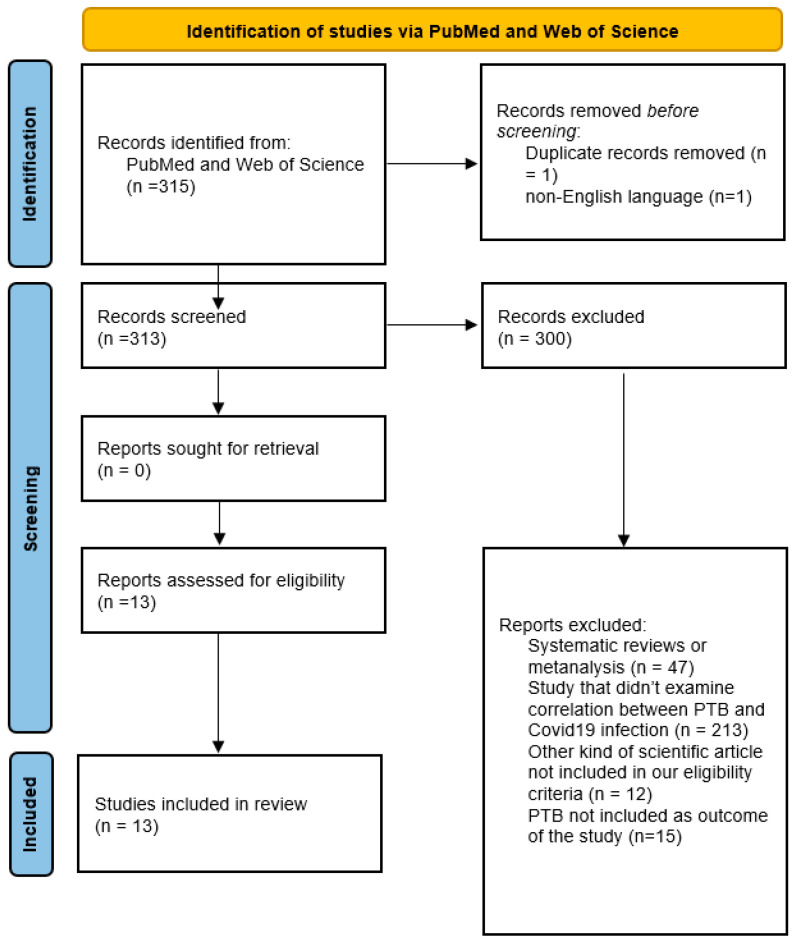
PRISMA 2020 flow diagram for new systematic reviews, which included searches of databases and registers only.

**Figure 2 biomedicines-13-00282-f002:**
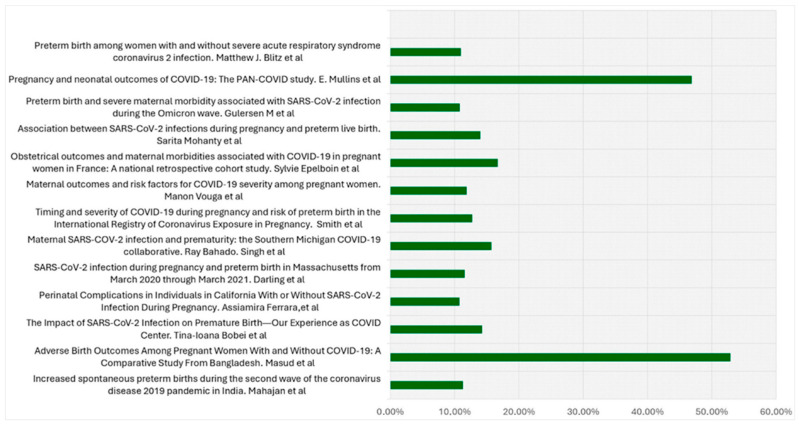
Preterm birth rates in pregnant women with COVID-19 in included studies [10,11,12,13,14,15,16,17,18,19,20,21,22].

**Table 1 biomedicines-13-00282-t001:** Collected Data.

Title, Authors	Country, Duration of Observation	Type of Study	Aim of this Study	COVID-19+ Patients	COVID-19− Patients	Preterm Birth in COVID-19+ Patients	Preterm Birth in COVID-19− Patients
Obstetrical outcomes and maternal morbidities associated with COVID-19 in pregnant women in France: A national retrospective cohort study—Epelboin et al. [10]	France. January to June 2020	Prospective Cohort Multicentric Study	Investigation on whether maternal morbidities were more frequent in pregnant women with COVID-19 diagnosis compared to pregnant women without COVID-19 diagnosis during the first wave of the COVID-19 pandemic.	874	243,771	146 (16.7%)	17,215 (7.1%)
Increased spontaneous preterm births during the second wave of the Coronavirus Disease 2019 pandemic in India—Mahajan et al. [11]	COVID-19 hospital Mumbai, India. 4 April 2020 and 4 July 2021.	Hospital-based, retrospective cohort study	To compare spontaneous preterm birth (SPTB) and iatrogenic preterm birth (IPTB) rates during both waves of the Coronavirus Disease 2019 (COVID-19) pandemic	1136	3463	128 (11.3%)	259 (13.8%)
Maternal SARS-COV-2 infection and prematurity: the Southern Michigan COVID-19 collaborative—Bahado-Singh et al. [12]	Michigan, USA, from March 2020 till 1 October 2020.	Multicentre case-control study	To determine the impact of COVID-19 disease on PTB overall and related subcategories such as early prematurity, spontaneous, medically indicated PTB, and preterm labor.	369	1090	58 (15.72%)	96 (8.81%)
Perinatal complications in individuals in California with or without SARS-CoV-2 infection during pregnancy—Ferrara et al. [13]	Northern California, 1 March 2020, and 16 March 2021.	Cohort study	To examine the risk of perinatal complications in pregnant individuals with SARS-CoV-2 infection.	1332	42,554	143 (10.74%)	3438 (8.08%)
Timing and severity of COVID-19 during pregnancy and risk of preterm birth in the International Registry of Coronavirus Exposure in Pregnancy—Smith et al. [14]	The USA. June 2020–July 2021	International internet-based retrospective cohort study	To estimate the risk of PTB (overall, spontaneous, and indicated) after COVID-19 during pregnancy while considering different levels of disease severity and timing.	1192	4692	152 (12.75%)	414 (8.82%)
Preterm birth is not associated with asymptomatic/mild SARS-CoV-2 infection per se: pre-pregnancy state is what matters—Cosma et al. [15]	Italy. 20 September 2020 and 9 January 2021.	Case-control study	To determine the real impact of asymptomatic/mild SARS-CoV-2 infection on PTB not due to maternal respiratory failure.	53	176	21 (39.62%)	81 (46.02%)
The impact of SARS-CoV-2 infection on premature birth—our experience as COVID-19 Center—Bobei et al. [16]	Romania. from March 2020 to June 2021	Prospective observational study in a COVID-19-only hospital	To determine the impact of SARS-CoV-2 infection on PTB pregnancies	238		33 (14.28%)	8.2%
SARS-CoV-2 infection during pregnancy and preterm birth in Massachusetts from March 2020 to March 2021—Darling et al. [17]	Massachusetts, from 1 March 2020 to 31 March 2021	Retrospective cohort study	To examine the association between SARS-CoV-2 infection and spontaneous and provider-initiated PTB and how timing of infection and race/ethnicity as a marker of structural inequality may modify this association	2195	66,076	254 (11.57%)	4655 (7.04%)
Adverse birth outcomes among pregnant women with and without COVID-19: a comparative study from Bangladesh—Masud et al. [18]	Bangladesh, from March to August 2020	Cross-sectional study	To compare birth outcomes related to COVID-19 between Bangladeshi pregnant women with and without COVID-19	70	140	37 (52.9%)	42 (30.0%)
Association between SARS-CoV-2 infections during pregnancy and preterm live birth—Mohanty et al. [19]	USA. August 2020–October 2021	Prospective cohort study.	To examine associations between mild or asymptomatic prenatal SARS-CoV-2 infection and preterm live birth	185	769	26 (14%)	97 (13%)
Maternal outcomes and risk factors for COVID-19 severity among pregnant women Vouga et al. [20]	COVI-Preg international registry; 24 March and 26 July 2020.	Retrospective comparative Monocentric	Insight into the maternal outcomes and risk factors associated with COVID-19 severity in pregnant women	926	107	110 (11.88%)	8%
Preterm birth among women with and without severe acute respiratory syndrome coronavirus 2 infection—Blitz et al. [21]	New York City and Long Island. March 2020 and June 2021	Retrospective cohort study	To establish potential risks determined by a COVID-19-positive pregnancy toward the mother and the newborn.	1261	30,289	138 (10.98%)	2140 (6.78%)
Preterm birth and severe maternal morbidity associated with SARS-CoV-2 infection during the Omicron wave—Gulersen M et al. [22]	Nwe Tork. 1 December 2021 and 7 February 2022.	Retrospective cohort study	To evaluate the risk of PTB and severe maternal morbidity (SMM) in pregnant patients with SARS-CoV-2 infection during the most recent wave of the COVID-19 pandemic.	631	4107	68 (10.8%)	329 (8.0%)

## Data Availability

The original data presented in this study are available on PubMed.

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
