# Peer review of "Preterm Birth and SARS-CoV-2: Does a Correlation Exist?"

_biomedicines, 2025, doi:10.3390/biomedicines13020282_

Round 1

Reviewer 1 Report

Comments and Suggestions for Authors

This review summarizes the relationship between SARS-CoV-2 virus infection and preterm birth.

1. The introduction does not discuss the academic contribution of this paper;

2. The keywords in the search of the literature only include preterm birth and Covid-19 infection and should include SARS-CoV-2 virus,” pregnancy,” etc.

3. In the second part of Materials and Methods, the research methodology is not clear.

4. Section 3 Results is just a compendium of existing research results and lacks an analysis of the strengths and weaknesses of these studies. There is a lack of critical analysis about the results.

5. Section 4 should add an in-depth discussion on the mechanism of how SARS-CoV-2 virus infection affects preterm labor.

6. The conclusions in Section 5 are too brief. It is suggested to add the outlook of future research on the relationship between SARS-CoV-2 infection and fertility.

Author Response

Response to Reviewer 1

Thank you very much for taking the time to review this manuscript. Please find the detailed responses below and the corresponding revisions and corrections in track changes in the re-submitted manuscript.

Comment 1. The introduction does not discuss the academic contribution of this paper.

Response 1. We appreciate your feedback regarding the academic contribution of our paper. We have now added a subsection in the Introduction that highlights the significance of our review in advancing the understanding of SARS-CoV-2's impact on pregnancy outcomes, particularly preterm birth, and its implications for clinical practice (line 58-62).

Comment 2. The keywords in the search of the literature only include “preterm birth” and “Covid-19 infection” and should include “SARS-CoV-2 virus,” “pregnancy,” etc.

Response 2. We acknowledge your observation about our keyword selection. The search strategy has been expanded to include additional relevant keywords such as “SARS-CoV-2 virus,” “pregnancy,” and “maternal health.” This will help to ensure a more comprehensive literature search (line 64-66).

Comment 3. In the second part of Materials and Methods, the research methodology is not clear.

Response 3. To enhance clarity, we have revised the Methods section to provide a more detailed description of our research methodology, including the inclusion and exclusion criteria, steps taken to assess study quality, and the approach for data extraction (line 71-88).

Comment 4. Section 3 Results is just a compendium of existing research results and lacks an analysis of the strengths and weaknesses of these studies. There is a lack of critical analysis about the results.

Response 4. We recognize the need for critical analysis in the Results section. We have now included a comparative analysis highlighting the strengths and weaknesses of the studies reviewed, discussing issues such as study design variability and potential biases, which should provide more context for our findings (line 253-281).

Comment 5. Section 4 should add an in-depth discussion on the mechanism of how SARS-CoV-2 virus infection affects preterm labor.

Response 5. An expanded discussion on the mechanisms by which SARS-CoV-2 may influence preterm labor has been added to Section 4. This includes the effects of inflammation, hypoxia, and other maternal physiological changes associated with COVID-19 (line 302-327).

Comment 6. The conclusions in Section 5 are too brief. It is suggested to add the outlook of future research on the relationship between SARS-CoV-2 infection and fertility.

Response 6. The conclusions have been elaborated to include an outlook on future research directions, focusing on potential longitudinal studies and investigations into the impact of emerging variants of the virus on preterm labor (line 369-386).

We believe that these revisions significantly enhance the manuscript's clarity, depth, and relevance. We thank the Reviewer for the constructive feedback, which has greatly contributed to improving our work. We look forward to your consideration of our revised manuscript.

Reviewer 2 Report

Comments and Suggestions for Authors

Thank you for the opportunity to review this manuscript.

The scope is suitable for Biomedicines but unfortunately the execution of this systematic review has some major errors.  

The manuscript presents a systematic review exploring the correlation between SARS-CoV-2 infection during pregnancy and the incidence of preterm birth (PTB).

the authors aim to highlight the increased risk of PTB associated with maternal COVID-19 infection.

Summarising the strength of this review:

The use of PRISMA guidelines and the inclusion criteria for studies enhance the credibility of the review.

- The authors have ensured the selection of diverse geographical and socioeconomic contexts

MAJOR ERRORS:

Only pubmed is used -this is not acceptable

A search was performed on PubMed utilizing the keywords "preterm birth" and "Covid-19 infection." =the search strategy should be broader

The search included all articles published from March 2020 to De-63 cember 2023 that fulfilled specific eligibility criteria

The biases are not discussed

A sensitivity analysis to assess the impact of smaller studies should be conducted

The review combines diverse study designs (retrospective cohort, case-control, etc.), which might introduce bias.

Author Response

Response to Reviewer 3

Thank you very much for taking the time to review this manuscript. Please find the detailed responses below and the corresponding revisions and corrections in track changes in the re-submitted manuscript.

Comments 1, 2.

1) Only pubmed is used -this is not acceptable.

2) A search was performed on PubMed utilizing the keywords "preterm birth" and "Covid-19 infection." =the search strategy should be broader.

Response 1, 2. We appreciate your suggestion for an expanded search strategy beyond PubMed. In response, we have now included Web of Science as additional database in our literature search process to enhance the comprehensiveness of our review. Moreover, the search strategy has been expanded to include additional relevant keywords such as “SARS-CoV-2 virus,” “pregnancy,” and “maternal health.” This will help to ensure a more comprehensive literature search (line 64-66).

Comment 3. The search included all articles published from March 2020 to December 2023 that fulfilled specific eligibility criteria.

Response 3. We interpreted this comment as a suggestion to clarify the specific eligibility criteria. To enhance clarity, we have revised the Methods section to provide a more detailed description of our research methodology, including the inclusion and exclusion criteria,

Comment 4. The biases are not discussed.

Response 4. In the revised manuscript, we have included a dedicated subsection that discusses potential biases associated with the included studies, such as selection bias and reporting bias, and how these may affect overall conclusions (line 206-252).

Comment 5. A sensitivity analysis to assess the impact of smaller studies should be conducted.

Response 5. We agree that a sensitivity analysis is essential. Although the data extraction from existing studies provides a comprehensive overview, we have included a statement indicating the challenges of conducting a formal sensitivity analysis due to the variability in study designs. We propose the need for this analysis in future research (line 241-246).

Comment 6. The review combines diverse study designs (retrospective cohort, case-control, etc.), which might introduce bias.

Response 6. We acknowledge the potential for bias introduced by combining diverse study designs. We now address this concern in both the Methods and Discussion sections, explaining our rationale for including studies of different designs and how this diversity can enrich our understanding while also mentioning the limitations related to heterogeneity (line 71-88 and 206-252).

Caption of figures: we modified the caption of figure 1; regarding figure 2, the number that appeared (-92) was probably due to formatting of page and it refers to the number of line.

We believe that these revisions significantly enhance the manuscript's clarity, depth, and relevance. We thank the Reviewer for the constructive feedback, which has greatly contributed to improving our work. We look forward to your consideration of our revised manuscript.

Round 2

Reviewer 1 Report

Comments and Suggestions for Authors

The paper has been revised.

Reviewer 2 Report

Comments and Suggestions for Authors

most comments have been addressed